# Forced choices reveal a trade-off between cognitive effort and physical pain

**Todd A Vogel¹\*, Zachary M Savelson², A Ross Otto¹†, Mathieu Roy¹†\***

¹Department of Psychology, McGill University, Montreal, Canada; ²Institute of Cognitive Science, Carleton University, Ottawa, Canada

**Abstract** Cognitive effort is described as aversive, and people will generally avoid it when possible. This aversion to effort is believed to arise from a cost–benefit analysis of the actions available. The comparison of cognitive effort against other primary aversive experiences, however, remains relatively unexplored. Here, we offered participants choices between performing a cognitively demanding task or experiencing thermal pain. We found that cognitive effort can be traded off for physical pain and that people generally avoid exerting high levels of cognitive effort. We also used computational modelling to examine the aversive subjective value of effort and its effects on response behaviours. Applying this model to decision times revealed asymmetric effects of effort and pain, suggesting that cognitive effort may not share the same basic influences on avoidance behaviour as more primary aversive stimuli such as physical pain.

## Introduction

People generally describe cognitive effort as an aversive experience to be avoided whenever possible (*Kurzban, 2016*). However, exerting effort is often instrumental in obtaining rewards, in which case the costs of effort must be weighed against potential gains and expected benefits of other, less effortful alternatives (*Kool and Botvinick, 2014*). Avoidance of effort, then, is thought to reflect a cost–benefit analysis aimed at optimizing cognitive resource allocation (*Kool and Botvinick, 2018*; *Kurzban et al., 2013*; *Shenhav et al., 2017*). However, at a phenomenological level effort feels intrinsically unpleasant beyond just an abstract 'cost'. Indeed, effort is often metaphorically associated with pain, as in familiar vernacular expressions such as 'it hurts to think' or 'my head hurts from thinking too hard'. However, direct comparisons between effort and pain remain to be examined beyond the mere metaphor. The goal of the present study was therefore to compare the aversive value of cognitive effort against that of physical pain through explicit choices whereby individuals must choose between the two aversive goods.

Effort and pain appear to share important similarities in terms of their respective functions; feelings of effort seem to promote conservation and optimization of limited cognitive resources (*Kool and Botvinick, 2018*; *Richter et al., 2016*), while pain is designed to trigger protective behaviours aimed at preventing injury (*Eccleston and Crombez, 1999*). In both cases, effort and pain appear to act as stop signals interfering with the pursuit of ongoing activities to preserve the energy and integrity of the organism (*Shackman et al., 2011*). Moreover, as with cognitive effort, responses to pain can be flexibly overridden in the service of greater rewards (*Vlaev et al., 2009*). However, wilfully accepting pain appears to require substantial self-regulatory effort, even in extreme situations where accepting a certain amount of pain could mean avoiding probable death (*Ralston, 2004*). Thus, an apparent conflict emerges between an instrumental system that can choose to accept pain in exchange for greater rewards, and a Pavlovian system commanding us to avoid injury at all costs (*Dayan and Seymour, 2009*). On this view, the drive to escape pain can override more deliberative behaviours so as to prevent us from overthinking our responses when injury is imminent (*Lewis et al., 2013*).

**\*For correspondence:**
todd.vogel@mail.mcgill.ca (TAV); mathieu.roy3@mcgill.ca (MR)

†These authors contributed equally to this work

**Competing interests:** The authors declare that no competing interests exist.

Interestingly, cognitive effort appears to share some of the aversive characteristics of pain: effort is generally felt as an unpleasant urge to disengage and avoid overly demanding actions. The origin of this unpleasantness and its effect on behaviour remains a topic of current discussion (*Shenhav et al., 2017*). Present leading theories define effort costs as the consequence of computational limits on cognitive systems, or as 'opportunity' costs that are later integrated in a cost–benefit analysis of the action (*Boureau et al., 2015*; *Kool and Botvinick, 2018*; *Kurzban et al., 2013*; *Shenhav et al., 2017*). Many of these studies estimate the 'subjective value' (SV) of effort by examining how rewards can be discounted by effort costs (e.g. *Chong et al., 2017*; *Westbrook et al., 2013*). Recent studies have also begun to highlight the mathematical (i.e. functional) form of cognitive effort costs and how the SV of effort changes non-linearly with increasing demands (*Białaszek et al., 2017*; *Chong et al., 2018*; *Chong et al., 2017*; *Kool and Botvinick, 2018*). The estimation of SV allows incommensurable goods, such as cognitive effort and pain, to be revalued as units on the same scale (*Levy and Glimcher, 2012*), allowing for better comparison and understanding of their influence on choices. However, past studies have traditionally defined SV in relation to other appetitive goods (e.g. monetary rewards; *Chong et al., 2017*; *Chong et al., 2018*; *Westbrook et al., 2013*). To avoid confusion, we refer to the *negative* SV of cognitive effort, or the aversive value that people attribute to increases in effort costs. As the aversive SV of effort cannot be observed directly, we leverage computational modelling to estimate its SV on the basis of choices made between effort and pain.

In turn, the relative aversiveness of cognitive effort and pain can be further examined through decision latencies (or response times; RTs). Using aversive SV as an analogue for effort costs, we can compare the speed at which choices are made, lending insight into the subjective valuations of cognitive effort and pain. For example, if effort and pain share similar aversive characteristics, we should expect higher aversiveness levels of a good to slow decisions made to that respective good— for example high pain levels should slow responses to choosing the pain option— indicating increased avoidance (e.g. *Kim et al., 2006*). At the same time, we should expect to see faster responses to choosing the alternative option—for example high effort levels speed responses to the pain option—indicating increased escape-like behaviour. Indeed, avoidance of pain involves a set of automatic processes that help us avoid further injury (*Andersen, 2007*), but these processes may be overridden by higher-order systems in the presence of competing goals (*Claes et al., 2015*; *Klein, 2015*; *Van Damme et al., 2012*). We therefore sought to examine whether avoidance of cognitive effort showed a similar pattern when the competing alternative was aversive, rather than rewarding.

In this study, we aimed at comparing the aversive properties of cognitive effort and pain to address the important but previously unexamined questions of (1) whether individuals will explicitly trade-off cognitive effort with physical pain, and (2) to what extent the subjective valuation of cognitive effort relates to pain and how this aversive SV subsequently affects decision latencies (i.e. RTs). If the first question is true, we can then make the striking prediction that people will accept physical pain to avoid exerting cognitive effort. To examine these questions, participants performed a simple economic decision-making task (e.g. *Westbrook et al., 2013*; *Westbrook et al., 2019*) where they had to choose between performing a cognitively demanding task or receiving painful thermal stimuli to avoid exerting effort.

By offering various levels of pain and cognitive demand—which were experienced directly after each choice—we were able to assess the 'pain value' of cognitive effort. We fit different computational forms of SV to participants' choices to rescale effort levels into units (SV) commensurable with pain. We then used the computed aversive SV of effort alongside pain levels to assess their impact on choice response latencies. That is, we analysed choice RTs to find potential similarities and differences in how cognitive effort and pain influence the speed with which decisions are taken.

Finally, as the decision to exert cognitive effort or experience pain is highly subjective, we were interested in inter-individual differences that govern how people value and choose to avoid effort or pain. While mental effort is generally described as aversive, a given level of effort is experienced as more costly for some than for others (*Cacioppo and Petty, 1982*; *Inzlicht et al., 2018*), suggesting that the intrinsic value of effort influences decisions to avoid it (e.g. *da Silva Castanheira et al., 2020*; *Sandra and Otto, 2018*; *Westbrook et al., 2013*). Similarly, there are well documented trait differences in how individuals evaluate and respond to pain (*Sullivan et al., 1995*). To this end, we were interested in exploring differences in traits directly relevant to aversion to effort expenditure

(*Cacioppo et al., 1984*) and pain (*Sullivan et al., 1995*) upon choices. However, decisions between cognitive effort and pain may be dependent on the actual experience of exerting effort or feeling pain, rather than the value attributed to them. Thus, we also individually calibrated task difficulty and pain intensity to control for differences in cognitive ability and pain perception, allowing us to better address the value attributed to pain or effort and its role in decision-making.

## Results

In a novel effort–pain decision-making paradigm (*Figure 1*), participants made a series of choices between a cognitively demanding task and a painful stimulus, each of which had five different levels that varied parametrically from low (i.e. low cognitive effort or low pain) to high (i.e. high cognitive effort or high pain). To elicit different levels of cognitive demand, we used the *N*-back working memory task, ranging from $N = 0$ to $N = 4$, wherein stimuli were sequentially presented, and participants asked to respond whether the stimulus presented $N$ times previous was the same or different from the current stimulus. The *N*-back task imposes increasing demands upon executive functions like selective attention, monitoring, and updating (*Smith and Jonides, 1999*), and is perceived as effortful, with self-reported effort increasing systematically with increasing *N* (*Westbrook et al., 2013*). Critically, we calibrated the difficulty of the *N*-back task (the interval between successive stimulus presentations; see *Buhle and Wager, 2010*) to account for differences in cognitive ability across participants (see Materials and methods). To elicit pain, thermal stimuli were applied to the participant's left forearm whereby increasing noxious heat would elicit exponential increases in reported pain intensity (*Atlas et al., 2014*; *Price et al., 1989*). In line with the *N*-back calibration, the painful temperatures used for each participant were also individually calibrated to account for subjective differences in pain perception (*Jepma et al., 2014*), in order to roughly equate the five levels of effort and pain across participants.

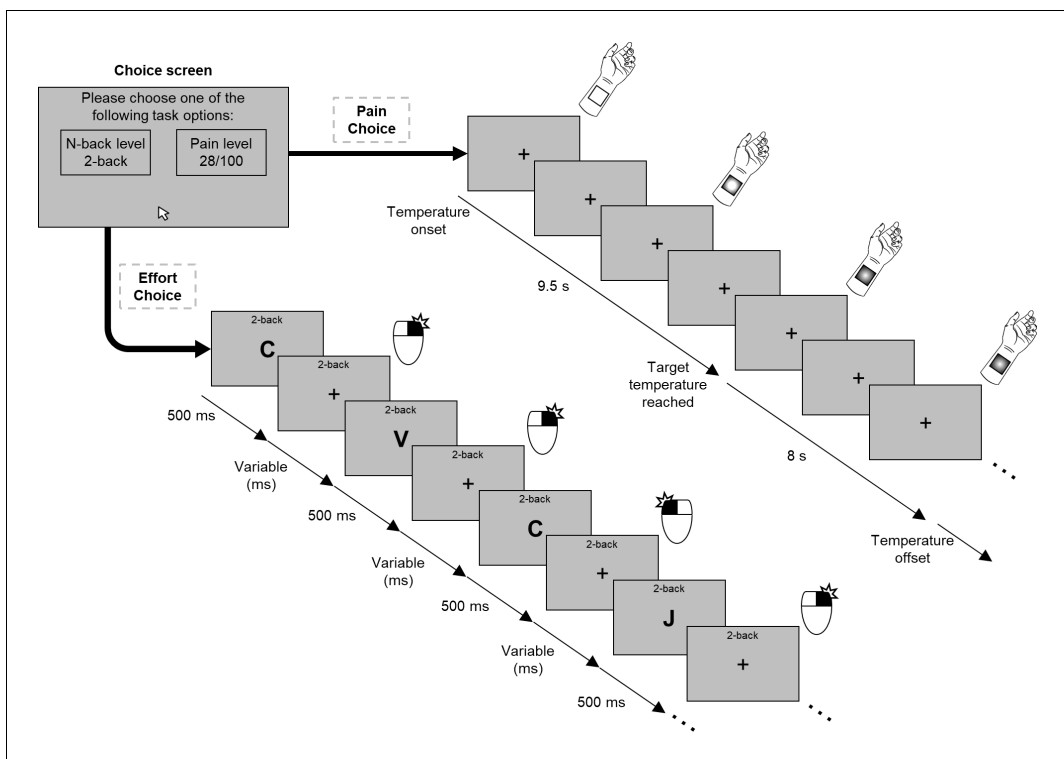

**Figure 1.** Example of a trial for the decision-making task. Participants made a choice between a level of the *N*-back task and a level of pain. If participants chose the effortful option, they performed the *N*-back task at the given level. If the painful option was chosen, participants received the corresponding painful stimulus and avoided performing the *N*-back task.

Following an established task design (*Soutschek et al., 2018*), we systematically paired each possible effort and pain levels to create 25 unique effort–pain pairs, each repeated twice. Depending on their choice, participants were either required to perform a trial of the offered *N*-back level or would receive the offered painful temperature. The duration of each decision outcome was equal (20 s) across all levels to ensure that participants' choices were not based on task length. As a manipulation check, we examined performance using the sensitivity measure *A* (a non-parametric alternative to *d'*; *Zhang and Mueller, 2005*; see Materials and methods) as a function of *N*-back level and pain ratings as a function of the calibrated temperatures, and found that performance declined with increasing *N*-back levels, $\beta_{N\text{-back level}} = -0.07$, 95% CI [−0.07,–0.06], $t(149.75) = 13.27$, p<0.001, and pain ratings increased with higher temperatures, $\beta_{Pain\ level} = 12.76$, 95% CI [11.50, 14.03], $t(111.61) = 19.56$, p<0.001 (see *Supplementary file 1A and 1B*). We also note that a pilot experiment conducted prior to the present study using a nearly identical protocol (*n* = 20) revealed qualitatively similar results to those described below (see Materials and methods).

## Choice behaviour

Examining choice behaviour, we found that at higher levels of cognitive demand people preferred to accept a physically painful stimulus rather than exert cognitive effort (see *Figure 2*). This perhaps surprising finding was supported, statistically, by a multilevel logistic regression predicting pain choices as a function of pain and *N*-back levels, which indicated a trade-off between cognitive effort and pain such that as the level of required cognitive effort increased, people were more likely to accept the painful stimulus in exchange for avoiding the effortful task, $\beta_{N\text{-back level}} = 1.60$, 95% CI [1.19, 2.04], $z = 7.34$, p<0.001.

Conversely, as the level of pain offered increased, people were more likely to choose the effortful cognitive task, $\beta_{Pain\ level} = -1.24$, 95% CI [−1.49,–1.02], $z = 10.40$, p<0.001. This observation reinforces the notion that cognitive effort is indeed aversive, even to the point where people sometimes prefer to feel pain rather than exert high levels of cognitive effort. We also observed a significant negative interaction between the level of cognitive effort and the level of pain offered, $\beta_{Pain\ level \times N\text{-}}$

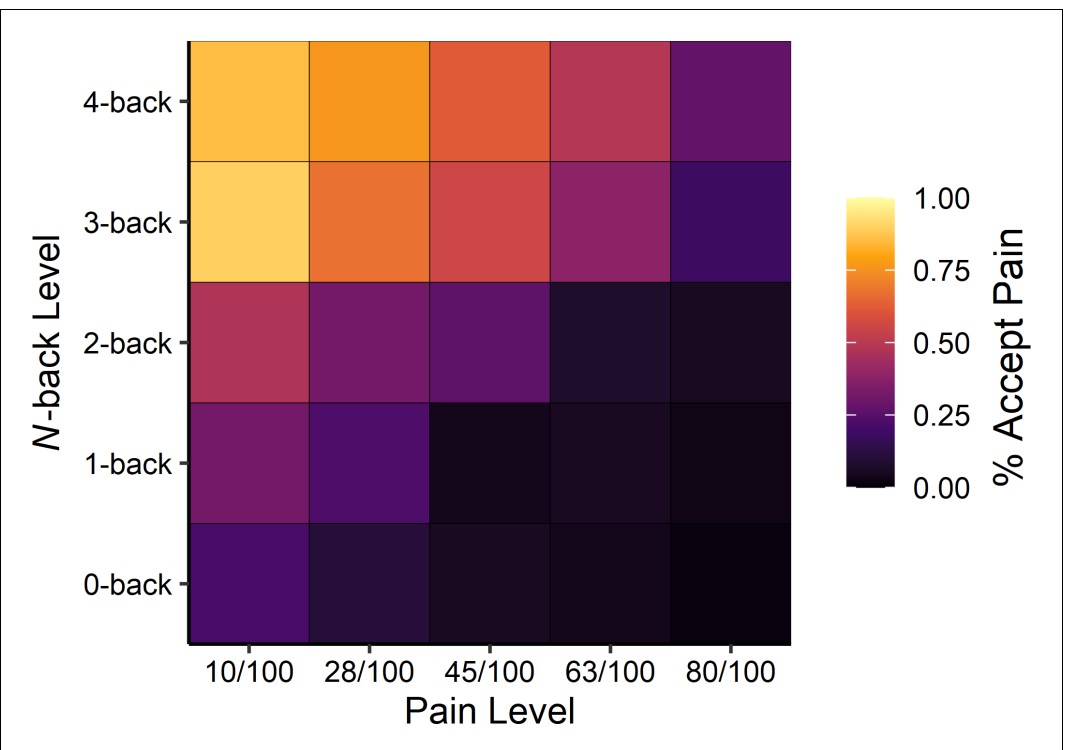

**Figure 2.** Averaged heatmap of choice behaviour across participants as a function of pain stimulus level (horizontal axis) and *N*-back level (vertical axis) on offer (see *Supplementary file 1C* for model estimates). Lighter colours indicate a higher percentage of accepting the painful stimulus.

$_{back\ level}$ = −0.12, 95% CI [−0.25, 0.003], $z$ = 2.03, p=0.042, suggesting that the scales governing effort and pain levels offered may not have been perfectly matched. This effect—which should be interpreted with caution due to the null value falling just inside the 95% confidence interval—suggests that at higher concurrent levels of pain and cognitive effort, there was a slight preference for choosing to exert cognitive effort over experiencing pain. Finally, adding trial number or the choice made on the immediately preceding trial did not significantly influence the results, suggesting that the slight preference for effort over pain was consistent over time (see *Supplementary file 1D and 1E*). Critically, average N-back performance was above chance at all levels of effort, demonstrating that participants were indeed engaged and correctly performing the task, $F$(1, 149.75)=175.98, p<0.001 (see *Supplementary file 1A*).

A separate intercept-only regression model examining the average probability of choosing the pain stimulus (not accounting for the level of pain or level of effort offered) showed that the painful option was chosen approximately 28% of the time, $β_{intercept}$ = −0.92, 95% CI [−1.25,–0.60], $z$ = 5.54, p<0.001. This overall difference in acceptance rates, however, does not necessarily imply a fundamental difference between cognitive effort and pain, but rather, that the levels we offered were not necessarily optimized for obtaining 50/50% acceptance/rejection rates.

## Computational modelling of aversive SV

To better understand the relationship between cognitive effort and pain, we fit a series of computational models to identify the mathematical form that best explained how the aversiveness of cognitive effort was subjectively valued in comparison to pain. Previous studies have used similar approaches to examine how exerting cognitive and physical effort discount potential monetary rewards (*Białaszek et al., 2017*; *Chong et al., 2018*; *Chong et al., 2017*; *Hartmann et al., 2013*), as well as the relationship between reward discounting by physical effort versus delay discounting (e.g. *Klein-Flügge et al., 2015*; *Prévost et al., 2010*). Here, rather than discounting rewards, we were interested in how increases in the aversive value (SV) of cognitive effort traded off with another well-known aversive experience: physical pain.

Following previous work on cognitive effort costs (*Chong et al., 2018*; *Chong et al., 2017*), we compared a series of models (see *Table 1*) fit to cognitive effort levels that assumed different functional forms of aversive SV—linear, exponential, parabolic, and hyperbolic—in order to transform cognitive effort into units more directly comparable to pain levels. As noted earlier, we calibrated the temperatures used for each participant based on their reported sensitivity to the painful stimuli and N-back speeds based on performance of the 2-back task. By assessing how each model accounted for each participant's trial-by-trial choices, we were able to capture how people value the aversive nature of cognitive effort. These four value functions were input to a logistic function predicting choices and fit using maximum likelihood estimation separately for each model (see Materials and methods). We found that participants' choices were best described by a parabolic weighting of effort levels (*Figure 3a*):

$$SV = k * E^2$$

where SV is the aversive SV of cognitive effort for a given level of effort E (0-back, 1-back, 2-back, 3-back, or 4-back), and k is a scaling parameter which signals the steepness of the function (higher values of k represent a larger sensitivity to the aversiveness of the effort level on offer). The newly computed SVs were then taken as input to a logistic choice function:

$$Pr(pain) = \frac{1}{1 + e^{β*(c+(P−SV))}}$$

Table 1. Goodness-of-fit Estimates from Computational Models on Cognitive Effort Levels.

|  | Linear | Exponential | Parabolic | Hyperbolic |
|---|---|---|---|---|
| AIC | 1223.32 | 1245.43 | 1170.66 | 1293.65 |

*Note.* Separate models were fit for each mathematical function on participants' data; AIC = Akaike Information Criterion (*Akaike, 1974*).

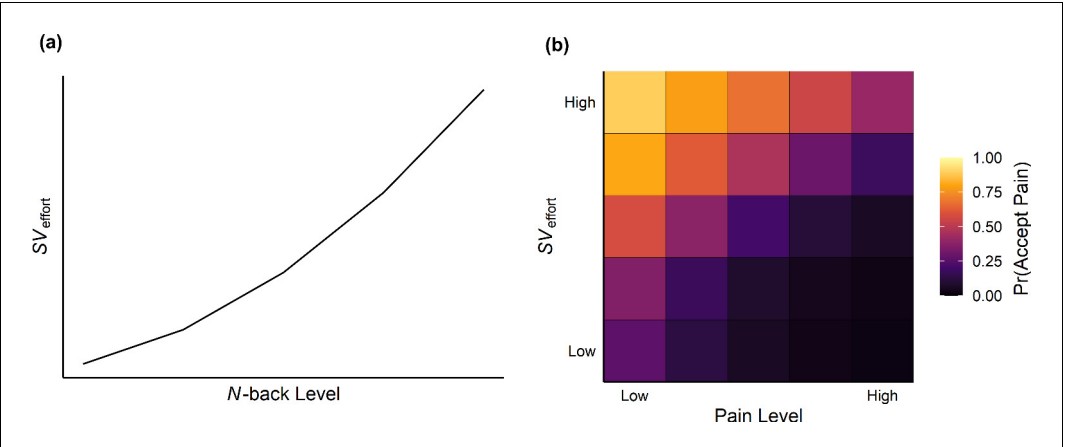

**Figure 3.** Parabolic scaling of effort levels and averaged heatmap of choices predicted from the computational model. (a) Higher offered effort levels led to a parabolic increase in the aversive SV of effort. This value was compared against the pain level on offer before being fit to participants' choices. (b) Averaged heatmap of choices predicted from the parabolic model fit to effort levels. Lighter colours indicate a higher probability of choosing the pain option.

where Pr(pain) represents the probability of choosing the pain option for a given SV against the pain level P on offer (10/100, 28/100, 45/100, 63/100, or 80/100; *Figure 3b*). The β parameter reflects the sensitivity of the logistic function to value differences, and the c parameter captures bias towards one option over the other (i.e. an overall preference for choosing pain over effort). Separate k (median = 0.18), β (median = 1.50), and c (median = 0.38) parameters were estimated for each participant.

## Choice RTs

Using the model-estimated SVs for cognitive effort, we examined choice RTs to choosing between pain and cognitive effort to see how the aversiveness of each good influenced the speed with which a choice was made. To do so, we estimated a multilevel linear regression to simultaneously predict RTs for both pain and effort choices, allowing us to separately examine the effects of the chosen and unchosen good upon choice latencies (see Materials and methods for full model specification). More simply, we used the effort and pain levels on offer to predict RTs to both options in a single regression model. Effort levels were transformed into aversive SV units by applying the parabolic function above using the median group-level *k* parameter for all participants.

Overall, there was a near-significant difference in RTs wherein participants took longer to choose the pain option as compared to the effort option (linear contrast, p=0.061; RT effect estimates are shown in *Figure 4* and *Supplementary file 1F*), supporting the notion that response behaviours can be inhibited by aversive stimuli (*Huys et al., 2011*; *Geurts et al., 2013*), and suggesting that the prospect of pain may exhibit a stronger inhibitory influence on choices than cognitive effort. There was also a significant effect of trial number showing that RTs became faster over time for both choices, *ps* < 0.022, but that RTs became slightly faster over time when choosing effort than when choosing pain (linear contrast, p=0.048).

Critically, for both choice types the aversiveness of the ultimately chosen option—that is $SV_{effort}$ when choosing the effort option, $\beta_{Choose\ Effort \times SVeffort}$ = 0.14, 95% CI [0.10, 0.19], $t(28.66)$ = 6.73, p<0.001, or pain level when choosing the pain option ($\beta_{Choose\ Pain \times Pain\ level}$ = 0.14, 95% CI [0.10, 0.18], $t(23.79)$ = 6.56, p<0.001—slowed RTs as the intensity of the offer increased). We observed no significant difference between the magnitude of these two effects (linear contrast, p=0.93), indicating that, for both cognitive effort and pain, increases in aversiveness slowed RTs when choosing the corresponding good (see *Supplementary file 1G & 1H* for averaged RTs predicted from the model and for correlations between predicted and observed RTs, respectively).

However, when examining how the aversive level of the unchosen good influenced RTs, we observed an asymmetry between effort and pain choices: when choosing the pain option, the

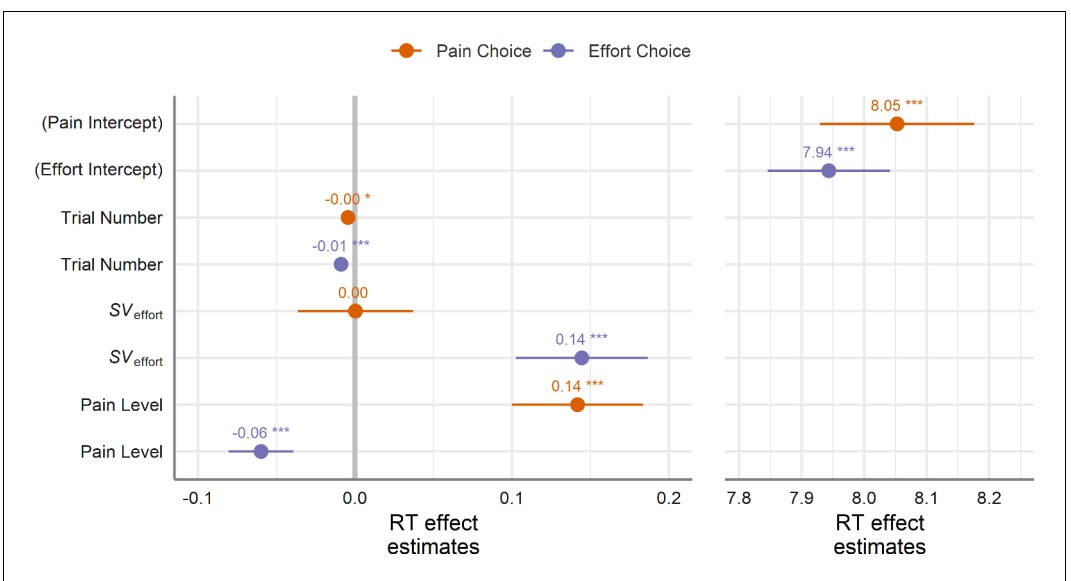

**Figure 4.** Estimated beta weights (presented in log-units) for the choice RT model. Positive values indicate an increase in predicted RTs (i.e. slowing down) and negative values indicate a decrease. RTs significantly increased at higher levels of the chosen good (i.e. at higher levels of $SV_{effort}$ when choosing the effort option and higher levels of pain when choosing the pain option). These effects remained even after controlling for the difficulty of the choices (*Supplementary file 1J*), controlling for a person's overall proportion of choices (*Supplementary file 1K*), and in those participants exhibiting an no overall bias towards one option over the other (*Supplementary file 1I*). Error bars represent the 95% CI based on the SEs from the model; \*\*\*p<0.001 (see *Supplementary file 1F* for full information on coefficient estimates).

aversive SV of effort had no significant effect on RTs, $\beta_{Choose\ Pain \times SVeffort}$ = 0.00, 95% CI [−0.04, 0.04], $t$(25.26) = 0.02, p=0.99; conversely, when choosing the effort option, the pain level on offer significantly sped RTs to choose the option, $\beta_{Choose\ Effort \times Pain\ level}$ = −0.06, 95% CI [−0.08,−0.04], $t$(27.81) = 5.63, p<0.001. A linear contrast revealed a significant difference between the magnitudes of these two effects (p=0.009), suggesting that pain had stronger speeding effects on choosing the alternative option (i.e. away from pain) than did cognitive effort.

While these findings suggest that avoiding pain has stronger effects on RTs than avoiding cognitive effort, it may be possible that these differences were driven by the overall imbalance in choices between pain and cognitive effort observed across participants. We conducted several supplementary analyses addressing the observed asymmetry in the effects of the unchosen option on RTs which converged on the same result: high levels of pain facilitated choosing the effort alternative, as demonstrated by reduced RTs.

As the observed asymmetry in RT may have been driven by those participants displaying an overall bias towards the effort option, or alternatively away from the pain option, we separately analysed those participants whose overall acceptance/rejection rate was not significantly different from 50% according to a logistic regression fit to each participant's choices; data from 11 participants met this criterion. This subset of participants, who exhibit no overall preference towards one option over the other, also showed that pain levels slowed RTs when choosing the pain option, and effort levels slowed RTs when choosing the effort option, $p$s < 0.001 (see *Supplementary file 1I* for model estimates). Examining the apparent asymmetric speeding effects, we found again that higher pain levels significantly sped RTs during effort choices (p=0.004), in line with our original findings, but no significant effect of higher effort levels on RTs for pain choices, p=0.36, suggesting that the observed asymmetry in the effects was not driven by an overall preference for the effort option in our sample.

We also performed an analysis that controlled for the difficulty of each choice (see *Supplementary file 1J*), in that pain levels appeared on the whole more aversive, and thus choosing effort over pain may have been generally easier and potentially faster. We found qualitatively similar results to our original model, wherein RTs slowed at more aversive levels of pain and effort to

choose the corresponding option, and that an asymmetry appeared when choosing the alternative option. That is, higher pain levels sped RTs to choosing the effort option, while higher SVs of effort appeared to slow RTs to the pain option (see *Supplementary file 1J*). A model examining the influence of a person's overall preference for effort or pain provided further support for our findings. Specifically, we included the proportion of pain choices for each person as a subject-level predictor and found similar RT effects for the aversive levels of the chosen and unchosen options (see *Supplementary file 1K*). Taken together, our RT analyses reveal that for both pain and effort, the aversiveness level of the chosen good slowed choices (*Kim et al., 2006*). However, a curious asymmetry emerged: the aversive SV of effort did not influence RTs for pain choices, but higher levels of pain sped RTs away from choosing pain and towards the alternative effort option.

### Individual differences in aversion to effort and pain sensitivity

The above findings indicate that people are willing to accept physical pain to avoid cognitive effort. However, an individual's propensity to accept pain—or effort—might also depend on how salient or aversive the two outcomes are to the individual decision-maker. For example, to some individuals the challenge of expending cognitive effort can be viewed as intrinsically rewarding (need for cognition [NFC]; *Cacioppo et al., 1984*; see also *Inzlicht et al., 2018*; *Sandra and Otto, 2018*). People also differ on the level at which pain is perceived as threatening (pain catastrophizing [PCS]; *Sullivan et al., 1995*). Below, we examine how individual differences in NFC and PCS bear upon on choice behaviour. However, given the sample size of the current study, we emphasize the preliminary, exploratory nature of these results.

First, we found that participants with higher-NFC levels were more likely to avoid pain and choose the cognitively effortful option when both goods were at higher intensities (see *Figure 5a*), possibly reflecting their higher willingness to exert cognitive effort (*Cacioppo and Petty, 1982*; *Westbrook et al., 2013*). Specifically, we found a significant moderating effect of NFC on the interaction between the level of pain and level of cognitive effort on offer, $\beta_{NFC \times Pain\ level \times N\text{-back}\ level} = -0.12$, 95% CI [$-0.25, -0.01$], $z = 2.12$, p=0.034, suggesting that as the levels of pain and cognitive effort increased, participants higher in NFC chose the cognitively effortful option more often compared to lower-NFC participants. Put another way, participants with higher-NFC levels appeared less sensitive to cognitive effort as pain levels increased relative to participants with lower-NFC levels. NFC did not significantly interact with any other effects in the model, $ps > 0.25$ (see *Supplementary file 1L*). Examining the moderating effect of NFC on *N*-back performance revealed no significant predictive effect, $\beta_{NFC \times N\text{-back}\ level} = 0.00$, 95% CI [$-0.01, 0.02$], $t = 0.92$, p=0.36.

Next, we found that pain catastrophizing also decreased acceptance of the painful option at higher intensities of effort (*Figure 5b*). More specifically, we found a significant moderating effect of PCS on the influence of the *N*-back level on choice behaviour, $\beta_{PCS \times N\text{-back}\ level} = -0.48$, 95% CI [$-0.87, -0.10$], $z = 2.42$, p=0.015, suggesting that participants with higher PCS were less sensitive to effort levels influencing their choice behaviour. Surprisingly, we did not find a significant moderating effect of PCS on any other predictors of choice behaviour, $ps > 0.31$ (see *Supplementary file 1M*). NFC and PCS scores were not significantly correlated, $r(36) = -0.26$, p=0.12, suggesting that the two traits separately influenced choice behaviour. Model estimates and inter-correlations between individual differences collected are reported in *Supplementary file 1N and 1O*.

## Discussion

How aversive is cognitive effort? Here using a forced-choice decision-making task between cognitive effort and pain we showed that cognitive effort can sometimes be aversive enough to drive people to accept physical pain as a means of escape. Perhaps surprisingly, as prospective cognitive demands increased, people were more likely to choose to receive a painful thermal stimulus rather than exerting effort. This provides compelling evidence that cognitive effort is aversive and the desire to avoid it can be quite strong. Put simply, people preferred to experience highly painful heat rather than do something mentally demanding. In fact, participants made effort-avoidant choices even when faced with the maximum ethically permissible pain level: on average participants accepted the painful stimulation calibrated to 80/100 (100 = 'extremely intense' pain) approximately 28% of the time to avoid performing the cognitively difficult 4-back task (see top-right corner of *Figure 2*). At the same time, as the level of pain offered increased, people were more inclined to avoid

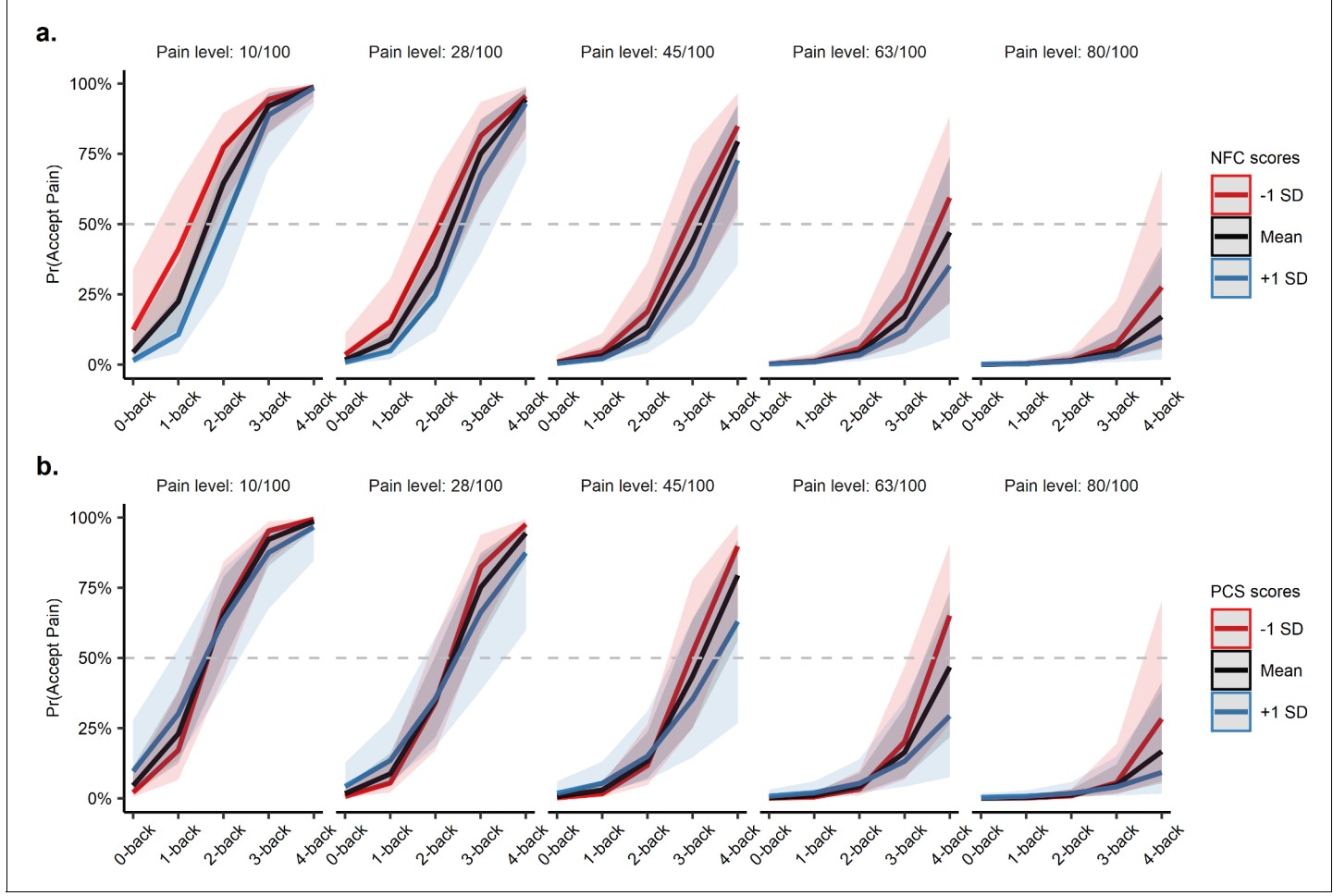

**Figure 5.** Predicted probabilities of accepting the pain option at each *N*-back and pain level moderated by need for cognition and pain catastrophizing. (**a**) Those higher in need for cognition were more likely to choose the effortful option, especially when pain and effort offers were both in the higher, more aversive range (e.g. 4-back vs. 80/100 pain). (**b**) Similarly, those higher in pain catastrophizing preferred to avoid the pain option (and thus choose the cognitive effort option) at increasing *N*-back levels. NFC = need for cognition, PCS = pain catastrophizing. The grey dashed line indicates the indifference point between the two choices (i.e. the point where the probability of choosing either the pain or effort option is 50%).

pain and instead choose to exert effort, demonstrating that, at some level, there is a trade-off between cognitive effort and primary aversive stimuli such as pain.

The trade-off between cognitive effort and pain suggests the presence of shared aversive characteristics between the two. Previous work examining choices between appetitive and aversive outcomes (*Plassmann et al., 2010*), as well as appetitive goods of different categories, has found that the choice options can be compared using a 'common currency' (*Chib et al., 2009*; *Levy and Glimcher, 2012*). It follows that aversive goods of different categories can be traded off once mapped to a common scale. To address this, we used computational modelling to infer the aversive SV of cognitive effort and rescale effort to units more directly comparable to pain. We found that the SV of cognitive effort increased parabolically when pain levels were assumed to be linear, showing that increases in the aversive value of effort were smaller at lower intensities of effort relative to higher intensities of effort. This finding contrasts with a recent study showing hyperbolic discounting of rewards by cognitive effort (*Chong et al., 2017*; *Chong et al., 2018*) and suggests that discounting by cognitive effort may be dependent on the category of the alternative good. This conclusion, however, remains to be further investigated by future studies examining discounting of different categories of goods by cognitive effort.

Informed by the modelling approach described above, we found that choices were slower for more aversive levels of the chosen good—i.e. higher $SV_{effort}$ levels predicted slower effort choices,

and higher pain levels predicted slower choices to pain. The aversiveness of cognitive effort and pain therefore appear to share similar inhibitory effects on actions, demonstrating that people will avoid cognitive effort in a similar manner to avoiding physical pain. However, we also observed that pain uniquely facilitated (i.e. sped) choice RTs to the opposing effort option, whereas offered cognitive effort did not significantly influence decision times to the alternative painful option. Several follow-up analyses suggested that this observed asymmetry did not appear to be driven by the imbalance in the levels of cognitive effort and pain that were offered.

These findings may suggest that as pain prospects increase, survival circuits designed to prevent injury and death may gradually exert more influence over decisions and drive decisions away from pain, even when the alternative may itself be aversive. Higher intensities of cognitive effort did not appear to exert this same level of control. That is, cognitive effort and pain are both avoided at higher intensities, but pain may additionally recruit more primitive avoidance systems that govern responses away from pain when the prospect of pain is high. Conversely, decisions to exert cognitive effort may solely rely on higher-order motivational systems (*Kurzban et al., 2013*; *Shenhav et al., 2017*) which do not engender the same level of urgency as more primary reinforcers, such as pain. However, as this study was the first to directly compare cognitive effort with physical pain, the generalizability of this observed asymmetry between pain and other aversive goods warrants further investigation.

Finally, we found that the decision between cognitive effort and pain was significantly influenced by trait differences in need for cognition (NFC) and pain catastrophizing (PCS). Higher NFC and higher PCS both predicted a preference for choosing cognitive effort over pain; however, the two measures were not correlated (see *Supplementary file 1O*) and thus appear to exert independent influences on choice behaviour. Specifically, as higher-NFC individuals perceive cognitive effort as less unpleasant (*Cacioppo and Petty, 1982*; *Cacioppo et al., 1984*), our results suggest that these individuals prefer effort over pain even when cognitive demands are highest (e.g. 4-back) or painful temperatures are lowest (e.g. 10/100, 0 = 'No pain at all'). This preference may reflect a higher intrinsic value for effort exertion or lower costs associated with it relative to lower-NFC individuals. At the same time, higher-PCS individuals tend to exaggerate, ruminate, and feel helpless about pain (*Sullivan et al., 1995*), and appear to avoid the painful option and choose effort even as the level of cognitive demand increases. Surprisingly, we found no relationship between PCS and the influence of pain levels on choice behaviour. That is, we did not find that higher-PCS individuals adjusted their decision behaviour at higher intensities of pain relative to lower-PCS individuals, but instead preferred to choose the alternative effortful option to avoid pain, even when that option itself may have been highly aversive. While these results suggest important trait differences in decisions between exerting effort and experiencing pain, we caution against over-interpretation of the findings. The current sample size was likely insufficient to obtain robust between-person effects, therefore we highlight these findings as preliminary. While it appears quite likely that inter-individual differences in perceptions about the aversiveness of pain or effort influence people's choices about pain and effort, the exact strength and form of these relationships remain to be delineated by future studies.

What, then, is the adaptive function of effort avoidance? Cognitive effort's intrinsic cost appears to prevent us from wasting processing resources and to disengage from inefficient actions, enabling those resources to be applied towards a different task or goal (*Inzlicht et al., 2018*; *Kool and Botvinick, 2018*; *Kurzban et al., 2013*; *Otto and Daw, 2019*). However, the urgency of the drive to avoid cognitive effort does not appear to share the same fundamental urgency as the drive to avoid further pain (*Navratilova and Porreca, 2014*; failure to avoid effort may be unpleasant, but failing to avoid pain can be life threatening; e.g. *Nagasako et al., 2003*). In effect, the purpose of treating cognitive effort as costly is to disengage us from those actions that are not sufficiently rewarding. By contrast, the ultimate function of pain is to preserve life and, as such, is the last line of defence against injury and further pain.

The findings reported above also lend insight into the role of cognitive effort in the experience and regulation of pain. The temperatures administered in the present study were above the activation threshold for nociceptors found within the skin (*Plaghki and Mouraux, 2003*; *Treede et al., 1995*) and show that the higher temperatures elicited a strong, painful sensation (max. temperature used: 49°C). Many studies have looked at the effects of cognition on acute and chronic pain (*Buhle and Wager, 2010*; *Bushnell et al., 2013*), but few, if any, have examined how the

aversiveness of cognitive effort may affect the pain experience and its potential role in making decisions about pain (**Bar-On Kalfon et al., 2016**).

It is important to note potential limitations of the forced-choice methodology we used here. First, we observed an imbalance in the choices people made between the two aversive options, suggesting that pre-selected levels of pain and effort were not perfectly matched. While this imbalance may have explained part of the asymmetry observed between pain and effort choices, follow-up analyses suggest that this was not the case. Findings from a balanced subset of the data and analyses controlling for relative differences in the SVs of effort and pain converged on the same findings of a trade-off between the two. Second, effort levels were calibrated by manipulating task difficulty and performance, which may not be linearly related to subjective effort. Therefore, the parabolic relationship observed here between pain and effort may have taken a different shape if effort levels were calibrated based on subjective ratings of effort. Future studies should aim at combining objective and subjective measures of effort to map out the relationship more clearly between pain and effort aversion.

In summary, these findings demonstrate that cognitive effort can be as aversive as physical pain—and, perhaps surprisingly, that people will sometimes prefer to feel physical pain than to exert cognitive effort. While these results highlight similarities in the aversive qualities of cognitive effort and pain, we also observed important differences. Contrary to cognitive effort, pain appears to elicit an avoidance response that interferes with instrumental decisions to prevent self-injury. This reflects the specialized role of pain in survival—address the source of (potential) injury or the consequences may be dire. Cognitive effort, although aversive, does not appear to have the same level of urgency: while thinking hard may 'hurt', it is unlikely to cause actual harm. A potential avenue for future research would be to leverage functional neuroimaging to understand how these two goods—cognitive effort costs and pain signals—could be encoded either by common or distinct regions, and how these regions direct avoidance behaviours, to further elucidate effort's similarities and differences with pain or other aversive physical experiences.

## Materials and methods

Thirty-nine healthy volunteers (29 females, mean age = 21.44 years, SD = 2.64 years) were recruited for the study through the McGill University community. Exclusion criteria for the study were previous or current diagnosis of a psychiatric or neurological disorder, chronic pain syndrome or neuropathy, history of alcohol or other substance abuse, and regular (>2 weekly) use of analgesics, anticonvulsants, narcotics, antidepressants, and/or anxiolytics. Participants were compensated for their time with either psychology course credits or at the rate of $15 CAD/hr. Informed written consent was obtained from all participants and the study was approved by the McGill University Research Ethics Board.

### Procedure

The study consisted of a single 2 hr session. After giving written informed consent, participants were given several computerized tasks to examine their choice behaviour between a painful thermal stimulus and a cognitively effortful task (the *N*-back task; see below). The E-Prime 3.0 software (**Psychology Software Tools, 2016**) was used to present the computerized tasks throughout the experiment. We first determined participants' sensitivity to the painful stimuli as well as their ability to perform the cognitive task. Following this, participants completed a series of questionnaires and measures of individual differences. Finally, participants were offered a series of choices between different levels of a painful stimulus and a difficult cognitive task. See below for the methods used and results of the pilot version of this study.

### Pain calibration

To determine a participant's sensitivity to the thermal stimuli, we applied several temperatures to the inner surface of the left forearm. For this, thermal heat was applied to four different sites on the surface of the arm using a 9 $cm^2$ thermal contact probe (TSA-II Neurosensory Analyzer, Medoc Ltd. Advanced Medical Systems, Israel). Five temperatures ranging equally between 45°C and 49°C were applied once to each of the four sites (20 total stimulations) from 32°C baseline to the target temperature (9.5 s rise time, 8 s plateau, 2.5 s fall time). Following each stimulation participants indicated

the sensation as either painful or non-painful and rated the sensation using a visual analogue scale (VAS). For temperatures rated as non-painful, participants rated the intensity of the sensation from 0 ('No warmth at all') to 100 ('Very hot, without pain'). If participants indicated the sensation as painful, they rated the intensity of the stimulus from 0 ('Not intense at all') to 100 ('Extremely intense'). These ratings were input into a regression model to calculate the temperatures used later in the experiment (*Jepma et al., 2014*; see *Supplementary file 1B* for average calibrated temperatures).

### N-back Calibration

After determining sensitivity to the thermal stimuli, and to adjust for differences in cognitive ability, participants completed a second calibration task performing the *N*-back working memory task. In this, letters were presented sequentially, and participants asked to indicate whether the currently presented letter was the *same* or *different* from the letter that was presented *N* letter characters previous. The 2-back version of the *N*-back was used to calibrate the inter-stimulus interval (ISI; the time between presented letters; min: 19 ms, max: 2583 ms) with each trial lasting 20 s. Before each letter, a fixation cross was presented for 250 ms, after which the letter character was presented for 500 ms. Participants performed 15 trials of the 2-back task with the ISI (i.e. difficulty) being adjusted after each trial to find a performance level between 0.75 and 0.85 based on the sensitivity measure *A* (*Zhang and Mueller, 2005*). *A* is a non-parametric alternative to the common sensitivity measure $d'$ that allows for hit or false alarm rates to be at or near 0 or 1, while also accounting for potential response bias. Perfect performance is indicated with a score of 1, chance performance with a score of 0.5, and a score of 0 if all responses are incorrect (i.e. responding opposite of instructions).

### Measures of individual differences

Following the calibration tasks participants completed several questionnaires as well as a measure of working memory capacity. The questionnaires administered included the Need for Cognition Scale (*Cacioppo et al., 1984*), the Pain Catastrophizing Scale (*Sullivan et al., 1995*), the Fear of Pain Questionnaire (*McNeil and Rainwater, 1998*), the Behavioural Inhibition System and Behavioural Activation System scales (*Carver and White, 1994*), and the Big-Five Inventory (*John et al., 1991*). Following completion of the questionnaires, participants completed a computerized version of the Operation Span task (*Unsworth et al., 2005*). See *Supplementary file 1O* for correlations between the different measures.

### Decision-making task

For the main experimental task, participants made a series of choices between five different levels of the *N*-back task (*N* = 0 to *N* = 4, where *N* = 0 required participants to respond to 'X' with one button and all other letters with a different button) and five different painful stimuli (temperatures corresponding to 10, 28, 45, 63, and 80 out of 100 on a pain scale). On each trial participants were presented with two options: one of the five options for the pain stimulus and one of the five options of the *N*-back. After making their decision, participants either performed the given level of the *N*-back or received the corresponding painful temperature. The duration of both the *N*-back and the painful temperature was 20 s to ensure that participants' choices were not influenced by the length of either option. The speed of the *N*-back (the presentation speed of the letters) was fixed across all five levels to the speed obtained from the calibration procedure at the beginning of the experiment (calibrated using the 2-back). Each level of the *N*-back was systematically paired with each level of pain and participants made a total of 50 choices between effort and pain (i.e. each effort–pain pairing was presented twice).

Accuracy feedback was provided after completing the *N*-back, but no punishments were given for poor performance (e.g. performing at or below chance level; see *Supplementary file 1A* for *N*-back performance findings). After receiving the painful temperature, participants rated the intensity of the stimulus on a Visual Analogue Scale (VAS) from 0 = 'No pain at all' to 100 = 'Extremely intense' so as to verify their perception of the painful stimulus (see *Supplementary file 1B* for pain ratings findings). After receiving feedback or providing ratings, there was a 2 s delay before the choice options for the next trial were shown.

## Data analysis

We constructed a series of multilevel models using the 'lme4' package (*Bates et al., 2015*) for the R language. To determine the random effects structure for each model, the maximal model was first fit (i.e. all random effects included) and then random effects systematically removed when the model failed to converge (e.g. *Barr et al., 2013*; *Matuschek et al., 2017*; see below for model formulae). Choice RTs were log-transformed to remove skew, and trials with (log-transformed) RTs outside three standard deviations from the participant's mean were removed from analysis (<1.2% of total trials). Data from one participant were removed from analysis due to failure to follow task instructions. Wald tests were used to evaluate the significance of the predictors in the logistic models on choice behaviour. Data from seven participants were excluded from analysis in the model on RTs due to a computer error in the starting position of the cursor on the screen. The Kenward–Roger (*Kenward and Roger, 1997*) approximation was used to test the significance of the predictors in the linear models on RTs, utilizing the 'doBy' package (*Højsgaard and Halekoh, 2018*) to linearly contrast the beta estimates where relevant. All significance tests were two-tailed and are supplemented with 95% bootstrapped confidence intervals (10,000 resamples) where possible. A null/intercept-only model on the response data revealed an intraclass correlation coefficient of. 21.

## Computational modelling

Participants' choices in the decision-making task described above were fitted with linear, exponential, parabolic, and hyperbolic functions to determine which functional form best represented the data (*Chong et al., 2018*; *Chong et al., 2017*). Data from one participant were not included in the computational modelling as the participant chose the effort option on every trial. Each function was separately fit to effort levels (i.e. *N*-back level) on offer during the task. The remaining data were input into a logistic function with maximum likelihood estimation using the *scipy.optimize.fmin* function in the SciPy package (*Virtanen et al., 2020*) for Python. The data and scripts used for computational modelling in Python can be found at https://osf.io/n4cht/. The mathematical functions fit to effort and pain levels were as follows:

$$\text{Linear}: SV = k * E$$

$$\text{Exponential}: SV = e^{k*E}$$

$$\text{Parabolic}: SV = k * E^2$$

$$\text{Hyperbolic}: SV = \frac{1}{1 - k * E}$$

where *SV* represents the aversive SV of cognitive effort, *E* represents the effort, or *N*-back, level on offer (coded 1, 2, 3, 4, or 5), *k* is the scaling parameter signalling the steepness of the function. The resulting SVs were then input to a logistic function:

$$Pr(pain) = \frac{1}{1 + e^{\beta * (c + (P - SV))}}$$

where *Pr(pain)* represents the probability of choosing the pain option, *P* represents the pain level on offer (coded 1, 2, 3, 4, or 5), $\beta$ is the parameter capturing the steepness of the logistic function, and *c* being a parameter to capture overall bias towards one of the two options. For each choice domain, separate *k*, $\beta$, and *c* parameters were used when fitting the model. Final model fits were compared using AIC, where AIC = −2*log(maximum likelihood) + 2 k, with *k* representing the number of parameters in the model (*Akaike, 1974*).

## Methods – Pilot experiment

A separate group of twenty healthy volunteers (15 females, mean age = 21.25 years, SD = 2.63 years) were recruited to participate in a pilot experiment similar to the study. The same exclusion criteria for the present study were used for the pilot experiment. Participants were compensated for their time and provided written informed consent as in the main experiment. The procedure for the

study was the same as the present experiment, except for some differences in the pain calibration procedure. Specifically, we used seven temperatures ranging from 40 to 49°C (40, 44, 45, 46, 47, 48, 49°C) to apply to the four sites. Temperatures rose quickly (2.5 s) from 32°C baseline, plateaued for 8 s, and fell quickly back to baseline (2.5 s). We later adapted, for the present experiment, the rise time and temperatures applied to better match that which is given to participants during the decision-making task. There were no other differences in methodology from the present experiment.

## Results – Pilot experiment

Similar to the findings on choice behaviour from the present experiment, we found that at higher levels of cognitive demand, participants preferred to accept the physically painful stimulus over exerting effort, $\beta_{N\text{-back level}}$ = 1.14, 95% CI [0.77, 1.55], z = 5.93, p<0.001. Additionally, we found an effect of pain level wherein participants were more likely to except the effortful cognitive task at higher levels of pain offered, $\beta_{Pain\ level}$ = −1.22, 95% CI [−1.65,–0.83], z = 6.00, p<0.001. Finally, we found a near-significant negative interaction between the level of cognitive effort and the level of pain offered, $\beta_{N\text{-back level}\times Pain\ level}$ = −0.11, 95% CI [−0.25, 0.02], z = 1.87, p=0.061.

Examining choice RTs revealed similar effects of pain level and effort level on choice behaviour to those found in the present experiment. We found no overall difference in RTs between the two options (linear contrast, p=0.39). Similar to the present experiment, we found that the level of aversiveness of the chosen option slowed RTs at increasing levels of aversiveness—i.e. N-back level for the effort option, $\beta_{Choose\ Effort\times N\text{-back level}}$ = 0.14, 95% CI [0.09, 0.19], t(18.74) = 5.75, p<0.001, and pain level for the pain option, $\beta_{Choose\ Pain\times Pain\ level}$=0.18, 95% CI [0.12, 0.23], t(17.29) = 6.11, p<0.001 (a linear contrast revealed no differences between the magnitude of these two effects, p=0.39). Additionally, for both choices there was a significant effect of trial number, ps < 0.001, wherein RTs became faster over time, but a linear contrast showed no significant difference between these effects (p=0.19).

We found similar effects to the present experiment on RT speeding: pain levels exerted a significant negative effect on RTs for effort choices, $\beta_{Choose\ Effort\times Pain\ level}$ = −0.06, 95% CI [−0.09,–0.02], t (18.48) = 3.09, p=0.006, and N-back levels exerted a non-significant effect on RTs for pain choices, $\beta_{Choose\ Pain\times N\text{-back level}}$ = −0.00, 95% CI [−0.08, 0.07], t(18.10) = 0.07, p=0.95. This asymmetry mirrored that found in present study, however, a linear contrast of these two effects did reach significance (p=0.23).

## Power analyses from results of pilot experiment

The *MLPowSim* software (http://www.bristol.ac.uk/cmm/software/mlpowsim/) was used to conduct a power analysis and compute sample size for the present study. Using estimates from the multilevel logistic regression on choice behaviour in the pilot experiment, and an alpha of 0.05, the analysis showed a sample size of 35 would be required to obtain at least 0.80 power for all three effects in the regression model.

## Analysis and visualization materials

The data and R scripts used for data analysis and visualization can be found at https://osf.io/n4cht/. Below is the R code used to fit the various regression models using the 'lme4' package (*Bates et al., 2015*). Trial-level variables (i.e. N-back level, pain level, $SV_{effort}$, trial number, and task choice) were level one in the models and participant-level variables (e.g. PCS, NFC) were level 2. 'nback_level', 'pain_level', and 'sv_effort' variables were mean centred; scores from individual difference measures were z-scored across participants. The overall choice behaviour model was fit using the glmer() function using the 'binomial' link and 'task_choice' being coded 0 = effort choice and 1 = pain choice:

$$task\_choice \sim nback\_level * pain\_level + (1 + nback\_level + pain\_level \mid subject)$$

Models taking individual difference measures as moderating variables were specified as follows:

$$task\_choice \sim nback\_level * pain\_level * nfc + (nback\_level + pain\_level \mid subject)$$

$$task\_choice \sim nback\_level * pain\_level * pcs + (nback\_level + pain\_level \mid subject)$$

To examine choice RTs, the lmer() function was used to estimate both effort and pain choices simultaneously:

$$log\_rt \sim 0 + pain\_choice + effort\_choice + pain\_choice:$$
$$(sv\_effort + pain\_level + trialnumber) + effort\_choice:$$
$$(sv\_effort + pain\_level + trialnumber) +$$
$$(0 + pain\_choice + effort\_choice) + pain\_choice:$$
$$(sv\_effort + pain\_level + trialnumber) +$$
$$effort\_choice:(sv\_effort + pain\_level + trialnumber \parallel subjectID)$$

## Acknowledgements

This work was supported by Natural Sciences and Engineering Research Council Discovery Grants awarded to ARO and MR, as well as a Fonds de recherche du Québec – Nature et technologies New University Researchers Start-up Program grant awarded to ARO.

## Additional information

### Funding

| Funder | Grant reference number | Author |
| --- | --- | --- |
| Natural Sciences and Engineering Research Council of Canada | RGPIN-2017-03918 | A Ross Otto |
| Fonds de Recherche du Québec - Nature et Technologies | 2018-NC-204806 | A Ross Otto |
| Natural Sciences and Engineering Research Council of Canada | RGPIN-2016-06682 | Mathieu Roy |

The funders had no role in study design, data collection and interpretation, or the decision to submit the work for publication.

### Author contributions

Todd A Vogel, Conceptualization, Formal analysis, Investigation, Visualization, Methodology, Writing - original draft, Writing - review and editing; Zachary M Savelson, Conceptualization, Investigation, Methodology; A Ross Otto, Conceptualization, Supervision, Funding acquisition, Methodology, Project administration, Writing - review and editing; Mathieu Roy, Conceptualization, Resources, Supervision, Funding acquisition, Methodology, Project administration, Writing - review and editing

### Author ORCIDs

Todd A Vogel (iD) https://orcid.org/0000-0003-0895-3845
A Ross Otto (iD) https://orcid.org/0000-0002-9997-1901

### Ethics

Human subjects: Informed written consent was obtained from all participants and the study was approved by the McGill University Research Ethics Board (REB File # 247-1117).

### Decision letter and Author response

Decision letter https://doi.org/10.7554/eLife.59410.sa1
Author response https://doi.org/10.7554/eLife.59410.sa2

## Additional files

### Supplementary files

• Supplementary file 1. This file contains the supplementary tables and figures referenced in the main text. (A) contains the average performance on the levels of the *N*-back task. (B) contains the

average pain ratings for the thermal stimuli. (C) contains model estimates from the multilevel logistic regression on choices. (D) contains model estimates from the multilevel logistic regression on choices after controlling for trial number. (E) contains model estimates from the multilevel logistic regression on choices after controlling for the choice made on the previous trial. (F) contains model estimates from the multilevel linear regression on choice response times (RTs). (G) contains the heatmaps of the average predicted RTs from the multilevel linear model. (H) contains scatter plots of predicted vs observed RTs for each combination of effort and pain levels. (I) contains the model estimates from multilevel linear regression on choice RTs for the subset of participants who displayed no significant bias towards either choice option. (J) contains the model estimates from the multilevel linear regression on choice RTs after controlling for the difficulty of the choices. (K) contains model estimates from the multilevel linear regression on choice RTs while including a subject-level predictor of a person's overall probability of choosing the pain option. (L) contains model estimates from the multilevel logistic regression examining the influence of need for cognition. (M) contains model estimates from the multilevel logistic regression examining the influence of pain catastrophizing. (N) contains model estimates from multilevel logistic regressions examining the influence of the other measures of individual differences. (O) contains correlations between the measures of individual differences.

- Transparent reporting form

### Data availability

All data analyzed for this study can be found on OSF (https://osf.io/n4cht/).

The following dataset was generated:

| Author(s) | Year | Dataset title | Dataset URL | Database and Identifier |
|---|---|---|---|---|
| Vogel TA, Otto R | 2020 | Forced Choices Reveal a Trade-Off between Cognitive Effort and Physical Pain | https://osf.io/n4cht/ | Open Science Framework, n4cht |

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
