## [Decision Letter]

**Acceptance summary:**

The reviewers and editors agreed that this is an important piece of work showing that, under some circumstances, humans will choose to experience physical pain rather than exerting cognitive effort (here, one round of an N-back working memory task). Interestingly, the authors find that the comparative costs of cognitive effort scale quadratically with working memory load, whereas pain scales linearly. The study design included some important controls, in particular ensuring that effort and pain levels were well-balanced to ensure maximal sensitivity in the tradeoff; and the authors conducted numerous careful analyses, using both choices and reaction times, to demonstrate that their results were not due to possible confounding factors.

**Decision letter after peer review:**

Thank you for submitting your article "Forced Choices Reveal a Trade-Off between Cognitive Effort and Physical Pain" for consideration by *eLife*. Your article has been reviewed by three peer reviewers, and the evaluation has been overseen by Jonathan Roiser as the Reviewing Editor and Christian Büchel as the Senior Editor. The following individuals involved in review of your submission have agreed to reveal their identity: Michael Inzlicht (Reviewer #1); Andrew Westbrook (Reviewer #2).

The reviewers have discussed the reviews with one another and the Reviewing Editor has drafted this decision to help you prepare a revised submission.

Summary:

This manuscript explores how human subjects trade different levels of cognitive effort (using N-back tasks for N up to 4) off against different intensities of thermal pain. The authors find that while most subjects choose to undertake cognitive effort rather than experience pain, there is a point at which subjects will choose to experience pain rather than complete one round of the n-back. They find that the comparative costs of the N-back scale quadratically with N, whereas thermal pain scales linearly; in a way that relates to individual differences in the need for cognition and pain catastrophizing. They also report some evidence based on reaction times for the choice that pain and effort are not compared on exactly the same scales.

This reviewers agreed that this manuscript considers an important question about the comparability of cognitive effort and pain, including whether effort engenders the same kinds of responses as pain, e.g. Pavlovian freezing, or escape behaviors. The question is timely and important. The study is unique in not only directly comparing the two forms of aversive conditions directly and allowing participants to trade them off against each other, but also in carefully ensuring that effort and pain levels are well-balanced to ensure maximal sensitivity in the tradeoff. Although there is substantial work on decision-making for pain and effort; this particular combination seems novel.

However, there were some substantial queries raised by the reviewers, as follows:

1) In relation to the between-person analyses, with only 39 participants (minus at least one after exclusions), the current study is simply not statistically powerful enough to make reasonable between-person comparisons given average effect sizes in psychology. This means that the null results for pain sensitivity might be false negatives; worse, it means that there is a decent chance that the positive associations with need for cognition could be false positives too. This is not necessarily the case here, but with such low power for between-subject analyses, it does not seem warranted to present the results as they currently are. It is recommend that the authors either do not present these results, or to present them with sufficient caveats, given the instability of such estimates with low power. To be clear, this critique only holds for the between-person analyses; the within-person analyses (the bulk of the paper) should be fine, though it is recommended that the authors mention power and how sensitive their design is to find effects of various sizes in the Materials and methods section.

2) What should we make of the fact that participants are not indifferent to pain vs effort? That is, averaging across all levels of pain and effort, people chose the painful option 28% of the time, preferring effort 72% of the time. The authors write, "This overall difference in acceptance rates, however, does not necessarily imply a fundamental difference between cognitive effort and pain, but rather, that the levels we offered were not necessarily optimized for obtaining 50/50% acceptance/rejection rates." Given this lack of balance, how should we interpret any of the differences between pain and effort? For example, the authors discuss the asymmetries between pain and effort at length. But how do we know these are real asymmetries and not simply the product of poorly calibrated pain levels? Is it possible that the SV of effort not influencing RTs when choosing pain reflects this poor calibration and not something inherent about pain? Similarly, how do we know that the longer RT for pain options is not a product of this poor calibration? The authors write, "Although we observed a trade-off between cognitive effort and pain, which suggests common aversive characteristics between the two, the present findings do not allow us to conclude that the aversive value of cognitive effort is necessarily equivalent to that of physical pain." The authors then describe all the reasons why these two modalities might be different; there is a similar statement in the Abstract. But, how can the authors be sure that this lack of equivalence is meaningful from their data set? This lack of calibration balance seems like a bigger problem than the authors make it out to be.

3) The authors make their biggest inference from an asymmetry in the effects of pain and load levels on RT. Specifically, they observe that people are overall slower to choose pain than effort and also that higher levels of pain uniquely sped effort choices, while higher levels of effort did not speed pain choices. Based on these two pieces of evidence, the authors argue that the two classes of aversive stimuli are distinct with pain driving more innate (e.g. Pavlovian) systems while effort deliberation is mediated by higher order systems (e.g. goal-directed SV comparison). Major concerns about this inference include:

a) If the authors wish to make inferences about choice RTs, it is critical to adequately control for differences in SVs as people are slower to decide when they are closer to indifference points. To control for distance from indifference in their RT analyses, they could consider including the (SVeffort – SVpain) difference as a regressor.

b) A difference in the effect of choice type on RTs might reflect that participants have a difference in what they treat as a "default" alternative. To test for this, the authors should try asking whether there were any asymmetries in the RT to choose effort when, according to their logit model, the p(choose effort) < 50% versus the RT to choose pain when p(choose pain) < 50%. The intuition is that they might be slower to override a default option when it has lower SV than it would be to override a non-default option when it has lower SV.

c) Also, the authors should test whether trial number interacts with select pain or select effort. A positive interaction for selection of one option but not the other would imply that people are increasingly relying on a default choice as decision fatigue sets in.

d) The authors have not acknowledged a fundamental difference between their pain and effort conditions regarding controllability. While selecting pain guarantees the experience of pain, selecting effort guarantees that participants will be invited to exert effort as they like, or not. At any point, participants could stop exerting effort and escape the effortful experience. By this logic, there may be no fundamental distinction between pain and effort, per se, but rather a fundamental distinction between controllable and uncontrollable aversive stimuli (which is well established, e.g. Seligman's work). It is not entirely clear what the authors could do to address this, aside from performing additional experiments in which they manipulate the controllability of the pain or effort condition (e.g. allowing people to actively escape pain at any moment). Short of that, or additional analyses the authors might consider with their existing dataset, the authors should list this as a major caveat on their inference in the Discussion.

4) If the Authors do continue to pursue the argument that slowing reflects a fundamental distinction between pain and effort, they should also consider other relevant literatures on this point. For example, if subjects slow to choose pain more than effort, does this mean that they are showing a kind of Pavlovian slowing the face of aversive cues (e.g. Meyer et al., 2019 PNAS), or does this mean that faster responding to choose effort mean that they are faster to respond when escaping pain? Relatedly, is the choice of effort versus pain an instance of active or passive fear response (and are there gender differences: e.g. Gruene et al., 2015 *eLife*)? Moreover, there is lots of great work articulating the differential effects of "response-competition, goal-directedness, and threat-imminence" on avoidance responses (see Cain, 2019 Current Opinion in Behavioral Sciences, for a nice review).

5) It would be useful to fit a model that considers sequential effects between successive trials as a function of what subjects chose. This is particularly important since pain may accumulate. Related to this point, what was the inter-trial interval, how was this recovery time determined? Equally, if subjects made mistakes on the N-back task on a trial, what effect did that have on subsequent choice(s)? It might be possible to use the repeated choices to study these effects.

6) The authors need to clarify what subjects were led to believe about the consequences of errors on the N-back. Could a form of risk sensitivity be affecting their choices?

7) The ultimate model of choice that the authors are advocating is confusing. Is the idea that there is some sort of two stage process in which pain takes some sort of non-cognitive priority; and only then is there a second process of “real” integration between pain and cognitive effort? Or perhaps a tertiary process by which subjects choose pain, and then reflect on whether this is really what they meant to do? The explanation requires substantial clarification.

8) The model of RTs was a bit confusing. Teasing apart the effect of the absolute difference in subjective value (which should speed up choice) and the effect that individually aversive choices (of high pain/effort), with the potential asymmetry between pain and effort in the latter, should be made transparent.

9) If the authors fit RTs by themselves, does the same parabolic function of cognitive effort that is inferred for choice fit best? And how closely related are the ks for individual subjects?

10) The result relating to the NFC scale is a bit strange. Figure 3A makes it look as if there is a main effect of NFC, at all levels of pain, the red lines are highest. However, the paper claims that NFC moderates the interaction between pain and effort. If these associations are retained in the paper (see point 1), this needs to be clarified

11) It wasn't always transparent how well the choice model recapitulated the various high order aspects of the behavior, burying the relevant relationships in a supplementary table seemed a bit unfortunate. For instance, how would Figure 3 look if generated from the model?

12) Similarly some of the figures do not provide as much understanding of individual differences in choice as they could. In Figure 2, for example, the format leaves no way of indicating how variable choice behavior was across subjects.

Essential revisions:

Based on these comments the following revisions and extra analyses are required if you choose to submit a revised manuscript:

1) Consider removing the individual differences results, given the sample size; or at least caveat them heavily explaining that they are preliminary and require replication (as per point 1 above)

2) Perform additional analyses including the (SVeffort – SVpain) difference as a covariate (as per point 3a above)

3) Check whether there were any asymmetries in the RT to choose effort when, according to their logit model, the p(choose effort) < 50% versus the RT to choose pain when p(choose pain) < 50% (as per point 3b above)

4) Test whether trial number interacts with select pain or select effort; or better, fit a model that considers sequential effects between successive trials as a function of what subjects chose (as per points 3c and 5 above)

5) Report the results of a model fitting RTs alone (as per point 9 above) and the extent to which the ks are consistent between this and the original model.

[Editors' note: further revisions were suggested prior to acceptance, as described below.]

Thank you for submitting your article "Forced Choices Reveal a Trade-Off between Cognitive Effort and Physical Pain" for consideration by *eLife*. Your article has been reviewed by two of the original peer reviewers, and the evaluation has been overseen by a Reviewing Editor and Christian Büchel as the Senior Editor. The following individual involved in review of your submission has agreed to reveal their identity: Andrew Westbrook (Reviewer #2).

The reviewers have discussed the reviews with one another and the Reviewing Editor has drafted this decision to help you prepare a revised submission.

Summary:

The reviewers were generally satisfied with your responses to their queries and remain of the opinion that the work is interesting and novel. They feel that the manuscript has been strengthened through including the extra analyses. However one point remains outstanding in relation to the "default" option, which requires some further analysis.

1) The reviewer apologises that, in the prior review, the meaning of "default" was evidently unclear. The basic idea is that a participant might have a tendency to choose one option more than another across trials. As a result, they might develop a kind of habit (or prejudice, or expectation of behavior), which would bias them towards one choice or another. In drift diffusion modeling terms, we might consider it to be a starting point bias. As such, overcoming that default bias requires additional evidence and time. Predictions arising from this idea would be that people are more likely to choose the default effort option (which the authors find) and slower to select the non-default pain option (which the authors also find).

However, it is also possible to consider how one alternative being a default might drive “asymmetric” interactions (which is the key piece of evidence motivating the claim that pain and effort have fundamentally different kinds of controllers). As such, it is critical to ensure that such a default bias is not the cause of the asymmetry.

Therefore, one control analysis that would be helpful would be to separately consider those participants who do not appear to have a default effort preference (i.e. those who choose effort <= 50%, and who are not slower to choose the pain) from those who do. Is there evidence of a similar asymmetric interaction among participants who do not show a default effort preference (at least qualitatively)? Is it different among participants who do ? Similarly, if the authors include a subject-level predictor (in a hierarchical regression model) as a covariate, the value of which is participants' average proportion of effort selection across all trials, is the asymmetry still evident? Is there an asymmetry, accounting for cross-level 3-way interactions?

---

## [Author Response]

Editor comment #1: Consider removing the individual differences results, given the sample size; or at least caveat them heavily explaining that they are preliminary and require replication (as per reviewers’ comment #1).Reviewers comment #1: In relation to the between-person analyses, with only 39 participants (minus at least one after exclusions), the current study is simply not statistically powerful enough to make reasonable between-person comparisons given average effect sizes in psychology. This means that the null results for pain sensitivity might be false negatives; worse, it means that there is a decent chance that the positive associations with need for cognition could be false positives too. This is not necessarily the case here, but with such low power for between-subject analyses, it does not seem warranted to present the results as they currently are. It is recommend that the authors either do not present these results, or to present them with sufficient caveats, given the instability of such estimates with low power. To be clear, this critique only holds for the between-person analyses; the within-person analyses (the bulk of the paper) should be fine, though it is recommended that the authors mention power and how sensitive their design is to find effects of various sizes in the Materials and methods section.

We agree with the reviewers that our sample size may not be ideal for detecting between-person effects and that our results should be presented with sufficient caveats and limitations. Given the relatively low sample size of the study for detecting these effects, we have revised the manuscript throughout to reflect the preliminary nature of the individual difference results. We opted to still present the results in the manuscript—with added caveats and limitations—rather than fully remove them to promote replication and exploration of the findings by future studies. Moreover, we had strong a priori hypotheses regarding PCS and NFC effects that aimed to reinforce the point that the trade-off between pain and effort may vary between individuals as a function of how they perceive the aversive value of pain and effort. The influence of these traits is likely, provided that we present our findings with the caveats associated with our sample size. From that perspective, we hope that the reviewers will be satisfied with our modifications. If not, we would be willing to remove the individual differences results completely if the reviewers feel strongly about these limitations.

Editor comment #2: Perform additional analyses including the (SVeffort – SVpain) difference as a covariate (as per reviewers’ comment #3a)Reviewers comment #3: The authors make their biggest inference from an asymmetry in the effects of pain and load levels on RT. Specifically, they observe that people are overall slower to choose pain than effort and also that higher levels of pain uniquely sped effort choices, while higher levels of effort did not speed pain choices. Based on these two pieces of evidence, the authors argue that the two classes of aversive stimuli are distinct with pain driving more innate (e.g. Pavlovian) systems while effort deliberation is mediated by higher order systems (e.g. goal-directed SV comparison). Major concerns about this inference include:Reviewers comment # 3a: If the authors wish to make inferences about choice RTs, it is critical to adequately control for differences in SVs as people are slower to decide when they are closer to indifference points. To control for distance from indifference in their RT analyses, they could consider including the (SVeffort – SVpain) difference as a regressor.

We agree with the reviewers that relative differences in SVs for pain and cognitive effort may influence RTs to choosing either option (indeed this is observed in several studies, e.g. De Martino et al., 2013, Nature Neuroscience). Additionally, we recognize that the observed imbalance in acceptance rates of pain/effort options raises concerns about the interpretability of the observed asymmetry in RT effects. To mitigate this imbalance and its potential effects on the estimability or interpretability of SV effects upon RT, we reanalyzed the data using two distinct approaches.

First, we reanalyzed a subset of the data that removed those trials with offers taken from the extremes (i.e. trials offering either the lowest level of effort, or the two highest levels of pain) to achieve an overall acceptance/rejection rate closer to 50% for the two choices and found qualitatively similar results to our original report. Specifically, we observed a trade-off wherein at higher levels of cognitive effort people were more likely to accept the pain stimulus to avoid exerting effort and at higher levels of pain more likely to choose the effort option. The intercept of the model was no longer significant, suggesting that acceptance rates were close to 50%. Moreover, the interaction between effort and pain levels was no longer significant, suggesting that the pain and effort level were better matched on subjective value. The findings from this reduced dataset have been added to the Choice RTs section of the Results. In brief, results from this “balanced” subset of offers completely replicated the results from the full dataset, suggesting that asymmetries between pain and effort choices may not be completely explainable by an imbalance in pre-selected levels of pain and effort.

Second, following the suggestion of the reviewers, we included a measure of distance from each subject’s computed indifference point by taking the absolute value of the difference between each participant’s predicted probability of choosing pain and chance probability (50%) of choosing pain. This analysis, which controlled for the difficulty of each decision, buttresses our initial findings of an asymmetry in RTs between pain and cognitive effort. We have included the new findings in the Choice RTs section of the Results. Again, this additional analysis also confirmed that the asymmetry between pain and effort cannot be explained by an imbalance in the proportion of “easy vs difficult” decisions for pain vs. effort choices.

In summary, we find that taking two distinct approaches—examining a subset of the data to effectually force more balanced choices, and controlling for decision difficulties—provide convergent evidence that our originally reported results are robust even in the face of imbalance choice proportions. That being said, we recognize that this is a potential issue, and have revised our Discussion to identify this choice imbalance as a potential limitation of the design taken in our work.

Editor comment #3: Check whether there were any asymmetries in the RT to choose effort when, according to their logit model, the p(choose effort) < 50% versus the RT to choose pain when p(choose pain) < 50% (as per reviewers’ comment #3b)Reviewers comment #3b: A difference in the effect of choice type on RTs might reflect that participants have a difference in what they treat as a "default" alternative. To test for this, the authors should try asking whether there were any asymmetries in the RT to choose effort when, according to their logit model, the p(choose effort) < 50% versus the RT to choose pain when p(choose pain) < 50%. The intuition is that they might be slower to override a default option when it has lower SV than it would be to override a non-default option when it has lower SV.

We agree that the relative values of pain and cognitive effort may play a key role in our observed asymmetries in the RTs (see our response to Editor comment #2 above). We are happy to test the suggestion, however, we must admit that we do not fully understand the reviewers’ comment. Here is our interpretation of their comment: When examining choice RTs, there is a “default” option in that the SV of one of the options is higher than the other. The option with the higher value would be considered the “default” option in this decision, and all else being equal, RTs choosing *away* from this default (i.e. higher-SV) option should be slower than RTs *toward* this option. Indeed, this is observed in a number of previous studies (e.g. De Martino et al., 2013; Konovalov and Krajbich, 2019).

Another interpretation of the reviewers’ comment is that if there is a default option, then RTs for the default vs. non-default option should be faster regardless of difference in SV between the two options. We believe that covarying out these differences with our “choice difficulty” parameter (see our response to Editor comment #2 above) might already address this question. Nevertheless, we performed the analysis below as suggested by the reviewers, which should help address our two interpretations of the reviewers’ comment.

In our RT regressions we included a new binary predictor that indicated whether P(choose pain) was above or below 50%—that is, when P(choose pain) < 50%, the predicted (i.e. “default”) choice for the trial would be the effort option. Likewise, when P(choose pain) > 50%, the default choice would be the pain option. This allowed us to contrast the overall RTs for choosing effort when P(choose pain) > 50% against the RTs for choosing pain when P(choose pain) < 50%. In other words, we contrasted RTs to choosing effort when the expected option was pain (i.e. going against the expected “default” option) and the RTs to choosing pain when the expected option was effort.

Author response table 1 provides a summary of the estimates from this model. The “Pain predicted” variable is a binary predictor where a value of 0 = P(choose pain < 50%) and a value of 1 = P(choose pain > 50%). In other words, when “Pain predicted” = 0, the effort option is expected to be chosen, and when “Pain predicted” = 1 the pain option is expected. We found no significant differences in RTs when pain was the expected option (i.e. the “Choose Pain × Pain predicted” and “Choose Effort × Pain predicted” terms; *p*s >.28). This may be a bit surprising as we would have expected RTs to be faster when people choose pain and pain is predicted vs. when effort is predicted (and vice-versa for effort). The absence of a significant effect here may be due to the small number of discordant trials where predicted and actual choices were not the same, although we can note that the regression weights are in the predicted direction. More crucially, a linear contrast between these two terms was tested to directly address the reviewer’s suspicion that there might be a “default” option, but the difference was not significant, suggesting that there was not a default option between the two choices (*p* =.89; note: the sign of the “Choose Pain × Pain predicted” term was flipped to test the effect when pain was not expected, i.e. P(choose pain) < 50%). Finally, our original RT effects of interest remained significant in this analysis. That is, higher levels of pain slowed RTs to choosing pain, while higher SVs of effort slowed RTs to choosing the effort option. There remained an asymmetry wherein pain levels sped RTs to choosing effort, but *SV*_effort_ had no significant effect on RTs to choose the pain option.

Thus, it does not appear that there is an asymmetry in RTs to choosing pain or choosing effort when a default option is being overridden. Due to our potential confusion surrounding the reviewers’ suggested analysis, and the apparent lack of effect, we have opted to omit this reanalysis in the manuscript. Of course, if we have misunderstood the reviewer’s comment, we would be happy to provide a response or report this re-analysis in the paper upon subsequent reviewer clarification.

**Author response table 1. resptable1:** 

Variable	β	95% CI	df	*t*	*p*
(Choose Pain Intercept)	8.13	[8.01, 8.24]	40.15	135.03	<.001 ***
(Choose Effort Intercept)	7.92	[7.82, 8.02]	31.99	164.95	<.001 ***
Trial Number	−0.01	[−0.01, −0.01]	31.23	7.19	<.001 ***
Choose Pain × Pain predicted	−0.06	[−0.18, 0.06]	35.83	1.09	.28
Choose Effort × Pain predicted	0.05	[−0.06, 0.17]	22.89	0.89	.38
Choose Pain × *SV*_effort_	0.02	[−0.02, 0.07]	28.17	0.99	.33
Choose Effort × *SV*_effort_	0.14	[0.10, 0.18]	25.86	6.58	<.001***
Choose Pain × Pain level	0.13	[0.08, 0.17]	25.78	5.74	<.001 ***
Choose Effort × Pain level	−0.06	[−0.08, −0.03]	30.63	4.91	<.001 ***

Editor comment #4: Test whether trial number interacts with select pain or select effort; or better, fit a model that considers sequential effects between successive trials as a function of what subjects chose (as per reviewers’ comments #3c and #5)Reviewers comment #3c: Also, the authors should test whether trial number interacts with select pain or select effort. A positive interaction for selection of one option but not the other would imply that people are increasingly relying on a default choice as decision fatigue sets in.Reviewers comment #5: It would be useful to fit a model that considers sequential effects between successive trials as a function of what subjects chose. This is particularly important since pain may accumulate. Related to this point, what was the inter-trial interval – how was this recovery time determined? Equally, if subjects made mistakes on the N-back task on a trial, what effect did that have on subsequent choice(s)? It might be possible to use the repeated choices to study these effects.

We thank the reviewers for the suggestion to examine potential learning effects on choices and RTs.

In response to reviewer comment #3c, we first tested whether trial number influenced the choices that people made by extending the multilevel logistic regression predicting choices and have included the findings in the Choice Behaviour section of the Results and Supplementary file 1E. Our main effects of interest remained significant, i.e. people were more likely to choose the pain option at higher levels of cognitive effort and less likely to choose the pain option at higher levels of pain. However, we found a significant interaction between trial number with effort level on offer, wherein people were less likely to choose the effort option at higher *N*-back levels over time. While we are reluctant to speculate in our report about this interaction reflecting a fatigue effect (especially considering the lack of observed *N*-back performance decrements over time, *p* =.12, or subjective fatigue ratings taken) we note that the direction of this interaction is compatible with the reviewers’ intuition. More importantly, we do not believe that these new results jeopardize any of our other interpretations and could be potentially interested in presenting them if the interpretation had been clearer. Finally, there was no significant interaction between trial number and pain levels, nor an interaction between pain and *N*-back levels.

Likewise, we reanalyzed our RT data by testing whether trial number interacted with select pain or select effort and have updated the model reported in the Choice RTs section of the Results, as well as Figure 4 and Supplementary file 1G, to reflect the findings from this new model. We found that, for both choice options, RTs became faster over time, with RTs becoming slightly faster over time when choosing effort than when choosing pain. After controlling for the effect of trial number, the effects of interest were qualitatively similar to our original report.

Next, in response to reviewer comment #5, we tested two models that accounted for sequential effects on choices and RTs, respectively. A multilevel logistic regression on choice behaviour, controlling for the choice made on the preceding trial, showed that people were less likely to choose the pain option if they had done so on the immediately previous trial, β_Prev. choice_ = −0.70, 95% CI [−1.13, −0.29], *z* = 3.27, *p* =.001. However, this effect was independent of the level of pain or cognitive effort currently on offer, *p*s >.18, suggesting that people were less likely to accept pain consecutively, regardless of the aversive levels on offer. This model has been added as Supplementary file 1F and referenced in the main text. We also tested whether *N*-back performance on the previous trial influenced subsequent choices, but found no significant effect of performance, *p* =.17.

A multilevel linear regression on RTs, accounting for the choice made on the previous trial (see Author response table 2), found that people were slower to choose pain if they had done so on the previous trial, β_Choose Pain×Prev. choice_ = 0.10, 95% CI [0.03, 0.18], *t*(1372.97) = 2.90, *p* =.004, but the previous choice had no significant influence on RTs to choosing effort, β_Choose Effort×Prev. choice_ = −0.02, 95% CI [−0.07, 0.03], *t*(1361.60) = 0.71, *p* =.48. After controlling for the choice made on the previous trial, the observed RT effects mirrored those originally reported. That is, RTs were slower when choosing effort at higher SVs of effort, *p* <.001, and slower when choosing pain at higher levels of pain, *p* <.001. There was no significant effect of SV of effort when choosing the pain option, *p* =.50, but pain levels significantly sped RTs to choose the effort option, *p* <.001.

Overall, we feel that controlling for learning effects (i.e. trial number) and sequential effects (i.e. previous choice) did not meaningfully change our results. We believe that participants were indeed making decisions based on the aversive levels of the current offers, and not on other factors such as time-on-task (see response to Editor comment # 13) or previous choices. We have updated the manuscript to reflect the findings from models including interactions with trial number and have described the findings from the model with previous trial choices here. Finally, the inter-trial interval used was 2 seconds to allow participants to prepare themselves before the next set of choice options appeared; we have added to the Decision-Making Task section of the Materials and methods this information.

**Author response table 2. resptable2:** 

Variable	β	95% CI	df	*t*	*p*
(Choose Pain Intercept)	8.06	[7.94, 8.18]	33.13	134.18	<.001 ***
(Choose Effort Intercept)	7.93	[7.84, 8.03]	32.75	161.52	<.001 ***
Trial Number	−0.01	[−0.01, −0.01]	31.13	7.18	<.001 ***
Choose Pain × Previous choice	0.10	[0.03, 0.18]	1363.83	2.91	.004 **
Choose Effort × Previous choice	−0.02	[−0.07, 0.03]	1343.70	0.71	.48
Choose Pain × *SV*_effort_	0.01	[−0.02, 0.05]	21.62	0.69	.50
Choose Effort × *SV*_effort_	0.14	[0.10, 0.19]	24.90	6.77	<.001 ***
Choose Pain × Pain level	0.14	[0.09, 0.18]	20.74	6.24	<.001 ***
Choose Effort × Pain level	−0.06	[−0.08, −0.04]	28.99	5.45	<.001 ***

Editor comment #5: Report the results of a model fitting RTs alone (as per reviewers’ comment #9) and the extent to which the ks are consistent between this and the original model.Reviewers comment #9: If the authors fit RTs by themselves, does the same parabolic function of cognitive effort that is inferred for choice fit best? And how closely related are the ks for individual subjects?

We thank the reviewers for the suggestion of modeling RTs directly to test the various functional forms of subjective effort valuation. We constructed a model to predict RTs from pain levels and *SV*_effort_ levels for both pain and effort choices, mirroring the logistic choice model. To examine the best-fitting functional form of effort, The *SV*_effort_ levels were transformed linearly, exponentially, parabolically, and hyperbolically, with respective *k* scaling parameters fit separately for each participant. These transformed *SV*_effort_ levels were then used to predict log-transformed RTs, using individual regression models, with the same predictor variables as the RT regressions reported in the main text.

Mirroring the choice model, we found that the parabolic function on effort levels had the lowest sum of squared residuals (i.e. error). There was a significant positive correlation between the participant *k* parameter values from the parabolic RT model and the *k* values derived from the choice model on choices, Kendall’s τ_b_ =.26, *p* =.023. These results suggest that a parabolic function best describes the subjective valuation of cognitive effort’s aversiveness, in comparison to other aversive stimuli like pain.

While we think the concordance between choice and RT models is potentially interesting, we hesitate to interpret our RT-based fits, and consequently highlight this finding in our revision as 1) previous literature examining subjective valuation of effort (e.g. Chong et al., 2017; Massar et al., 2020 Frontiers in Human Neuroscience; McGuigan et al., 2019, Brain) have only fit subjective value functions to choices (not RTs), and 2) as the focus of this work is not on subjective value function identification, we believe this analysis would be more distracting than helpful in the main text. Of course, if the reviewer feels strongly about its inclusion or mention, we would be happy to include this analysis upon future revision.

[Editors' note: further revisions were suggested prior to acceptance, as described below.]

Revisions for this paper:1) The reviewer apologises that, in the prior review, the meaning of "default" was evidently unclear. The basic idea is that a participant might have a tendency to choose one option more than another across trials. As a result, they might develop a kind of habit (or prejudice, or expectation of behavior), which would bias them towards one choice or another. In drift diffusion modeling terms, we might consider it to be a starting point bias. As such, overcoming that default bias requires additional evidence and time. Predictions arising from this idea would be that people are more likely to choose the default effort option (which the authors find) and slower to select the non-default pain option (which the authors also find).However, it is also possible to consider how one alternative being a default might drive “asymmetric” interactions (which is the key piece of evidence motivating the claim that pain and effort have fundamentally different kinds of controllers). As such, it is critical to ensure that such a default bias is not the cause of the asymmetry.Therefore, one control analysis that would be helpful would be to separately consider those participants who do not appear to have a default effort preference (i.e. those who choose effort <= 50%, and who are not slower to choose the pain) from those who do. Is there evidence of a similar asymmetric interaction among participants who do not show a default effort preference (at least qualitatively)? Is it different among participants who do ? Similarly, if the authors include a subject-level predictor (in a hierarchical regression model) as a covariate, the value of which is participants' average proportion of effort selection across all trials, is the asymmetry still evident? Is there an asymmetry, accounting for cross-level 3-way interactions?

We thank the reviewer for their clarification of a default bias. Following the reviewer’s suggestions, we conducted several supplementary analyses, described below, that converged on the same findings: high levels of pain facilitate choosing the effort alternative, as demonstrated by reduced RTs, but high levels of effort do not facilitate choosing the pain alternative.

As suggested by the reviewer, we first separately examined the RT data for participants whose overall likelihood of choosing the effort option was < 50% (i.e. there was a preference towards choosing the pain option). Only 5 participants from our original analysis met this criterion. Nevertheless, we conducted an analysis of RTs and found that pain levels slowed RTs to choosing the pain option (*p* =.039) and effort levels slowed RTs to choosing the effort option (*p* =.084), in line with our original findings. Examining the asymmetric interaction, we found that higher pain levels significantly sped RTs to choosing effort, β_Choose Effort×Pain level_ = −0.08, 95% CI [−0.14, −0.02], *t*(220.83) = 2.47, *p* =.015, but that there was no significant speeding of RTs to choosing pain at higher effort levels, β_Choose Pain×*SV*effort_ = −0.04, 95% CI [−0.15, 0.07], *t*(4.02) = 0.71, *p* =.52.

However, the likelihood of choosing the effort option may not have been *significantly* lower than 50% for all 5 participants. Using this stricter criterion, we found one participant for whom P(choose effort) was significantly lower than 50%, who also displayed a similar effect. That is, pain levels and effort levels significantly slowed RTs to choosing pain and effort, *p*s <.001, respectively. Pain levels significantly sped RTs to choosing the effort option, β_Choose Effort×Pain level_ = −0.07, 95% CI [−0.09, −0.04], *t*(17.67) = 5.06, *p* <.001, while effort levels had no significant effect on RTs to choosing the pain option, β_Choose Pain×*SV*effort_ = 0.01, 95% CI [−0.03, 0.05], *t*(815.25) = 0.56, *p* =.57.

The small sample size used here, however, warrants caution against over-interpretation of this analysis. To help address this, we separately analyzed those participants whose overall P(choose pain) was not significantly different from 50% according to a binomial regression fit to each participant’s choices; data from 11 participants met this criterion (4 came from the first analysis mentioned above). This subset of participants, which represent those who displayed no overall bias towards one option over the other, also showed that pain levels slowed RTs to choosing the pain option and effort levels slowed RTs to choosing the effort option, *p*s <.001. Examining the asymmetric effects, we found significant speeding of RTs to choosing the effort option at higher levels of pain, *p* =.004, but no significant effect of higher effort levels on RTs to choosing the pain option, *p* =.36. We have added the findings from this analysis in the Choice RTs section of the Results and Supplementary file 1I.

Second, we analyzed RTs while including a subject-level predictor of participants’ average proportion of pain selection across trials. We included this *Choice Proportion* variable as an overall covariate for each choice (centered on 0.5, higher value=more pain choices) and found a significant overall effect when selecting the pain option, β_Choose Pain×Choice Prop._ = −0.70, 95% CI [−1.36, −0.05], *t*(31.02) = 2.07, *p* =.047, but no significant effect when choosing the effort option, β_Choose Effort×Choice Prop._ = 0.45, 95% CI [−0.10, 1.00], *t*(30.16) = 1.63, *p* =.11. Most importantly, the asymmetry in RTs remained after controlling for a person’s overall proportion of pain choices, β_Choose Effort×Pain level_ = −0.06, 95% CI [−0.08, −0.04], *t*(27.85) = 5.70, *p* <.001, β_Choose Pain×*SV*effort_ = −0.00, 95% CI [−0.04, 0.04], *t*(24.65) = 0.15, *p* =.88.

Finally, we conducted a model that included the *Choice Proportion* variable as a moderator on each effect of interest. Examining the 3-way interactions, we found a significant positive interaction between effort levels and choice proportion on RTs to choosing the effort option, *p* =.032, and a near-significant negative interaction for the pain option, *p* =.090. Interestingly, there was no significant interaction between pain levels and choice preference on RTs to choosing the pain option, *p* =.46, nor on RTs to choosing the effort option, *p* =.80, suggesting that the asymmetric effects of pain on effort choice RTs and effort on pain choice RTs were not affected by participants’ preference for pain or effort. We have added the findings from this analysis in the Choice RTs section of the Results as well as Supplementary file 1K.

Altogether, these analyses converge to show that the observed asymmetry in RTs, i.e. that pain levels speed RTs to the effort option and effort levels have little effect on RTs to the pain option, does not appear to be dependent on a default choice towards the effort option. In all of our supplementary analyses, pain levels consistently sped RTs towards the effort option, even in those who did not display a bias towards one option over the other.